# Individual determinants of satisfaction with the work environment after relocation to activity-based workplaces: A prospective study

**Katarina Wijk** [1,2,3]*, **Eva L. Bergsten**[2], **Svend Erik Mathiassen**[2], **David M. Hallman**[2]

**1** Centre for Research and Development, Region Gävleborg/Uppsala University, Gävle, Sweden,
**2** Department of Occupational Health Sciences and Psychology, Faculty of Health and Occupational Studies, University of Gävle, Gävle, Sweden, **3** Department of Public Health and Caring Sciences, Uppsala University, Uppsala, Sweden

* Katarina.wijk@regiongavleborg.se

**Data Availability Statement:** All relevant data are within the manuscript and the Supporting file. For ethical reasons and in accordance with Ethical approve, age are not included in the supporting file

## Abstract

Relocation to activity-based workplaces influences work environment satisfaction, but individual determinants of changes in satisfaction remain unknown. The aim of the present study was to determine whether age, gender, education, occupational position, or office type before relocation can predict work environment satisfaction among employees and managers relocated to activity-based offices. Respondents (n = 422) rated work environment satisfaction three months before and nine months after relocation. The findings indicate that, on average, satisfaction decreased after relocation, while for some workers it increased. Occupational position and office type at baseline predicted changes in satisfaction with the work environment; specifically, managers and those working in open-plan offices before relocation reported a smaller decline in satisfaction after relocation, compared to those relocating from private offices. Participants with no university education were more satisfied with the physical and psychosocial work environment in activity-based workplaces than those with a university degree.

## 1. Introduction

Working life has become more flexible, more digitalized, and more interactive, and this has introduced new ways of communicating and interacting among employees [1]. Consequently, there is a need to adapt office workplaces to these new ways of working [2].

One way of doing this is to change the office space, for example by relocation to activity-based workplaces. In activity-based workplaces, employees share non-assigned desks and various areas designed for flexibility, e.g. open meeting spaces, enclosed rooms, and spaces allowing different types of activities such as interaction with colleagues or silent concentration [3, 4]. Personal devices are usually removed upon leaving the desk and are stored in toolboxes, and certain office rules are made explicit, including a clean desk policy and that interaction between

as it then might be possible to identify individuals. Information on age at the individual level is available on a reasonable request to the first author.

**Funding:** This research was funded by the Swedish Transport Administration www.trafikverket.se (DH, EB, KW), Region Gävleborg www.regiongavleborg.se (KW) and The University of Gävle www.hig.se (DH, EB), Sweden. The funders had no role in study design, data collection and analysis, decision to publish, or preparation of the manuscript.

**Competing interests:** The authors have declared that no competing interests exist.

colleagues needs to take place in specific areas. One reason for the increasing interest in activity-based offices is that less office space is needed, which may reduce costs for employers. Thus, the workplace design, and therefore the physical and psychosocial work environment, is changed in activity-based workplaces. These changes might also lead to changes in employees' behavior [5], satisfaction, and perceived productivity [4]. Both positive and negative psychosocial outcomes related to the work environment have been demonstrated in a review on new ways of working, e.g. in activity-based workplaces [6].

## 1.1 Activity-based workplaces

Activity-based workplaces have been widely adopted, and for some workers, health and work satisfaction can be affected, i.e. the sense of coherence (manageability, meaningfulness, and comprehensibility) for working in an activity-based workplace [7, 8]. Mache and Servaty [9] found reduced mental demands and stress but increased workload after relocation to activity-based workplaces. Personal preferences might affect how activity-based offices are perceived [10]. In a systematic review, Engelen and Chau [11] found that for some individuals, working in activity-based offices influenced health, work performance, and perceptions of the working environment. Positive outcomes were found for communication, social interaction, work time control, and satisfaction with the workspace, but evidence for effects on mental and physical health was limited. Other studies show that productivity and job satisfaction might be influenced by new ways of working, such as in flexible workplaces [12]. Satisfaction with the physical work environment, communication, and privacy is associated with productivity and well-being at work; for instance, time lost while looking for a desk at an activity-based workplace can have a negative effect on productivity and well-being [13]. In another study on new ways of working, the physical work environment was investigated and the authors found that it predicted work engagement [14]. Perceived physical and psychosocial working conditions [10] and health [15–18] are affected by activity-based workplaces. Overall, research indicates both negative and positive changes in work environment satisfaction when relocating workers to activity-based workplaces. This suggests that individual factors may play an important role in how activity-based workplaces are perceived, but previous studies do not identify which individual factors–e.g. age, gender, education, job position, and office type at baseline–predict work environment satisfaction in such workplaces.

## 1.2 Individual and structural determinants

A theoretical model of risks and benefits associated with working in activity-based offices suggests that well-being, satisfaction, and motivation might be moderated by tasks, personality, and organizational factors [19, 20]. The interplay between individual determinants and structural factors such as the work environment might depend on age, personal values, and job rewards [21].

From previous studies, it is known that factors such as age and individual preferences, e.g. need for privacy, relate to perceived fit in activity-based work [3, 22]. Job satisfaction is associated with age in both traditional [23] and activity-based workplaces [3, 22]. Motivation and satisfaction in younger individuals increase as they are offered career opportunities, whereas older individuals are more in need of intrinsic challenge and fulfilling work [23]. Job satisfaction tends to improve with age but also tends to decrease the longer workers stay at a particular job [24]. This might indicate that workers' perception of the work environment may change with age, but little is known about whether this is true even for activity-based work environments. Mauno et al. [25] found that younger individuals in the service sector reported higher job satisfaction than older colleagues, and previous research also reveals age-related differences

in distraction during performance of an auditory task [26]. It is therefore interesting to study age in connection with work environment, where some people relocate from cell offices to working environments that could be perceived as including more distractions if there are more people sharing rooms.

Gender differences in satisfaction with open workspaces have been found, with men feeling less positively about them than women [27]. Similar differences were also found for satisfaction in terms of access to supportive facilities. These results may mean that sharing workspaces and facilities is perceived less positively by men than by women, while teamwork, conversely, is more important to women due to the risk of conflicts. This may possibly be due to gender differences in interpersonal strategies, which in turn are associated with the design of the office in which the employees work. Previous studies also highlight gender differences in workplace conflicts related to office types, where conflicts for women were particularly related to noise in the office [28].

To our knowledge, there is a lack of research on the extent to which educational level and work satisfaction are associated in activity-based workplaces, beyond the fact that social background determines various outcomes at work [29], and that education determines job satisfaction in general [30].

Furthermore, satisfaction with activity-based workplaces, is likely related to user expectations [31]. This highlights the importance of considering the office type before relocation, since expectations can contribute to the perception of the activity-based workplace [20, 32].

Given that a growing body of literature emphasizes the influence of activity-based workplaces on work satisfaction in general, and the importance of individual factors in particular, we find it useful to further examine the extent to which different individual factors have an impact on work satisfaction in activity-based workplaces.

Thus, our aim was to determine the extent to which individual factors, i.e. age, gender, education, job position, and office type at baseline, predict satisfaction after relocation to activity-based workplaces with (I) workplace design, (II) the physical work environment, and (III) the psychosocial work environment.

## 2. Materials and methods

The current prospective study was conducted at two regional office sites of a large governmental agency in Sweden. A relocation to activity-based workplaces was carried out during the summer vacation in August 2018 at one regional office, and six months later in January 2019 at the other regional office. The relocation involved employees and managers who worked in either a private or shared office, or in an open-plan office.

The agency initiated the relocation and contacted our team of researchers to evaluate its implementation. To our knowledge, no other major structural changes or interventions were planned or implemented within the organization during the relocation.

Written consent was obtained electronically, before answering a questionnaire. Participants confirmed via email that they had been informed about and agreed to participate in the study. The Regional Ethical Review Board in Uppsala, Sweden approved the study (Ref. no. 2015/ 118).

### 2.1 Participants

Table 1 shows the response rates. In total, 1,063 people were invited to participate. We initially included all employees working at the agency who were expected to relocate to new premises. Three months before relocation (defined as baseline), 698 responded, and 422 responded both at this occasion and again at follow-up nine months after relocation. Criteria for exclusion

**Table 1. Respondents at baseline and follow-up.**

| Questionnaire distributed | Invited to participate | Respondents Baseline n (%) | Respondents Follow-up n (%) |
|---|---|---|---|
| Office site A | 288 | 215 (75%) | 151 (52%) |
| Office site B | 775 | 483 (62%) | 271 (35%) |
| Overall response rate | 1,063 | 698 (66%) | 422 (40%) |

were employees being on sick leave or parental leave, or reporting career changes or retirement in advance. Employees who did not relocate or had been assigned customized workplaces were also excluded, meaning that all participants in the study, regardless of their position or tasks, worked in activity-based workplaces after relocation. Tasks undertaken at the agency were mainly administrative, although the exact profession of employees varied. The non-assigned area in the new office, designed for flexibility and different types of activity, was divided into zones labelled e.g. as quiet zones or meeting rooms. Before the relocation, a designated project group ensured that the activity-based workplace would be dimensioned to fit the needs of everyone. During the working day, a given worker was supposed to change zones depending on the task, moving to the zone specifically designed for the current task. Before relocation, a number of education and information initiatives were carried out in order to ensure that everyone was aware of the rules applicable at the new office. A detailed description of the relocation process and office parameters before and after relocation has been published previously [33].

## 2.2 Measures and analysis

A comprehensive web-based questionnaire was used to collect data on individual characteristics and satisfaction with the work environment three months before (baseline) and nine months after (follow-up) relocation to activity-based offices. The questionnaire was constructed and administered using Webropol (www.webropol.com) and sent to eligible participants via an email containing an invitation to participate and a personal link to the questionnaire. A first reminder was sent one week after the initial invitation, and then another two to three weeks later.

Before relocation, the office type was allocated to one of two categories. In a "private or shared office", one to four people share one cell office with a door that can be closed, while an "open-plan office" is larger and shared by many employees. In both types of office, workers had their own designated workspace.

We used work environment satisfaction, focusing on perceived satisfaction with the physical and psychosocial work environment, as a measure of the overall perception of the work environment. Satisfaction regarding workplace design was measured using six questions. The dependent variables were, "How satisfied are you regarding. . ." (a) your opportunities to reach any one of your colleagues quickly for short meetings; (b) physical distance between you and your colleagues; (c) visual privacy at work; (d) the size of your workspace in terms of interaction with visitors; (e) the design of the workplace; and (f) the availability of technical tools to carry out your work. The responses were given on a scale ranging from 1 (very dissatisfied) to 7 (very satisfied). An index was created by averaging the score for all items relating to workplace design.

Satisfaction with the physical and psychosocial work environment was measured using two items: "Regarding your work in general, how satisfied are you with the physical work environment?", and likewise for the psychosocial environment, with a response scale ranging from 1 (very dissatisfied) to 5 (very satisfied). The questionnaire also contained validated questions on health issues [34–36].

Independent variables were: age, "What is your age (type your age)"; job position (employee/manager), reference employee; education, "What is your highest level of education" (elementary school/upper secondary school/vocational training/university degree); gender (male/female/other), reference female; and office type at baseline (private or shared office "reference cell-office" or open-plan office). Education was categorized as either no university education or university education, reference no university education. Respondents answering "other" on the gender question (n = 3) were excluded, as they were too few for analyzing separately, and confidentiality could not be guaranteed due to the limited number.

## 2.3 Statistical analyses

All statistical analyses were conducted in SPSS 25.0. Baseline and follow-up data were described using means and standard deviation (SD) for continuous variables and frequencies and percentages for categorical variables. A scatter plot was also made in order to describe how respondents ranked satisfaction regarding workplace design and physical and psychosocial work environment before and after relocation. This was described with Spearman's rank correlation.

Normal distributions were tested for in SPSS. After confirmation of satisfying normal properties, a change score was calculated for the satisfaction variables by subtracting the baseline value from the follow-up value. This resulted in three dependent variables indicating the change in satisfaction with workplace design, physical work environment, and psychosocial work environment after relocation to activity-based offices. Linear regression analysis was then used to determine the extent to which the individual factors age, gender, education, position, and office type at baseline predict satisfaction with the relocation. First, univariate models were constructed using each of the predictors (age, gender, education, position, and office type) as independent variables and the change in satisfaction as the dependent variables. This was done separately for satisfaction with workplace design, physical work environment, and psychosocial work environment. Second, three multivariate regression models were conducted (one for each satisfaction variable) by entering all predictors in the model simultaneously. In each model, we determined the explained variance (adjusted $R^2$) and the beta coefficients with 95% confidence intervals. Significance level was set to $p < 0.05$.

## 3. Results

The study sample of 422 respondents who responded at both baseline and follow-up did not differ significantly from those responding only at baseline (n = 698) in terms of the proportion of women or those in a management position (p>0.05). The study sample was slightly older, had a higher level of education, and more frequently in a private/shared office at baseline (p<0.05).

Descriptive data of the total study sample (n = 422) stratified by office type before the relocation to activity-based offices are presented in Table 2. The workers in open-plan offices were slightly older, included slightly more women, and were less satisfied with the work environment before relocation. The proportion of workers with higher education and a management position did not differ markedly between those located in open-plan offices and those in private/shared offices.

## 3.1 Satisfaction with workplace design

**3.1.1 Satisfaction with workplace design before and after relocation.**   The descriptive analysis showed that on average, respondents were less satisfied with the office design nine months after relocation to activity-based workplaces compared to before. Fig 1 shows that for

**Table 2. Descriptive data of the total study sample and those located in private/shared offices or open-plan offices before relocation to activity-based offices.**

|  | Total sample (n = 422) | Private/shared office (n = 274) | Open-plan office (n = 131) |
|---|---|---|---|
| **Age**, mean (SD) | 46.5 (9.8) | 44.4 (8.4) | 48.0 (9.6) |
| **Gender**, n (%) |  |  |  |
| Women | 197 (46.7) | 120 (43.8) | 65 (49.6) |
| Men | 222 (52.6) | 152 (55.5) | 65 (49.6) |
| Other* | 3 (0.7) | - | - |
| **Education**, n (%) |  |  |  |
| Elementary school, secondary school or vocational training | 139 (32.9-) | 94 (34.3) | 39 (30.8) |
| University degree | 283 (67.1) | 180 (65.7) | 92 (70.2) |
| **Position**, n (%) |  |  |  |
| Employees | 388 (91.9) | 254 (92.7) | 118 (90.1) |
| Managers | 34 (8.1) | 20 (7.3) | 13 (9.9) |
| **Office type**, n (%) |  |  |  |
| Private or shared office | 274 (67.7) | - | - |
| Open-plan office | 131 (32.3) | - | - |
| **Satisfaction**, mean (SD) |  |  |  |
| Physical work environment (scale 1–5) | 3.9 (1.0) | 4,2 (0.8) | 3.3 (1.0) |
| Psychosocial work environment (scale 1–5) | 3.9 (0.9) | 4.0 (0.9) | 3.7 (1.0) |
| Workplace design (scale 1–7) | 5.1 (1.3) | 5.6 (1.1) | 4.3 (1.1) |

* Excluded in order to ensure anonymity.

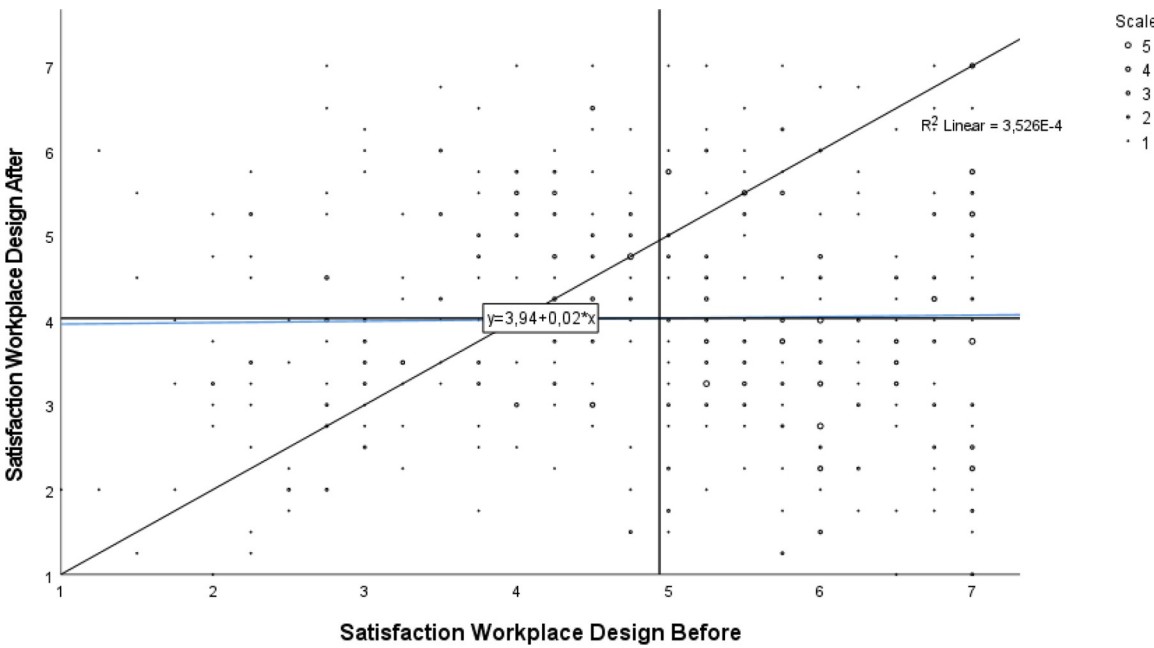

**Fig 1. Satisfaction regarding workplace design before and after relocation, index based on six variables.** N = 422. Note: The vertical line shows the group mean at baseline, and the horizontal line shows the group mean at follow-up. Line of identity is included in the diagram, as well as the regression between satisfaction before and after the relocation.

**Table 3. Satisfaction regarding workplace design in the activity-based office, compared to before the relocation.** Univariate linear regressions, n = 422.

|  | B (95% CI) | Sig. | Adjusted $R^2$ |
|---|---|---|---|
| **Age**: *Years* | -0.034 (-0.054 to -0.015) | **0.001** | 0.025 |
| **Gender** *ref female* | -0.356 (-0.746 to 0.033) | 0.073 | 0.05 |
| **Education** *ref not university* | 0.030 (-0.384 to 0.443) | 0.887 | -0.002 |
| **Position** *ref employee* | 1.231 (0.527 to 1.936) | **0.001** | 0.025 |
| **Office before relocation** *ref cell office* | 1.848 (1.469 to 2.226) | **<0.001** | 0.184 |

Note: The dependent variable is the difference in the index for satisfaction regarding office design, nine months after relocation compared to three months before relocation.

most respondents, perceptions of workplace design changed during relocation. However, for some the satisfaction increased and for others it decreased, as shown by the low correlations (Spearman's rank correlation 0.017), and we did not find any association between ratings before and after relocation.

**3.1.2 Univariate associations between individual factors and change in satisfaction with workplace design after relocation.** Age was negatively associated with the change in satisfaction with workplace design (B = -0.034, p = 0.001), i.e. older employees perceived a larger decrease in satisfaction with workplace design than younger employees (Table 3). A significant association was also shown for occupational position (B = 1.034, p = 0.001): managers were less dissatisfied with changes in workplace design compared to employees. Office type before relocation also predicted satisfaction with workplace design (B = 1.848, p<0.001): individuals who had been in an open-plan office at baseline before relocation were less dissatisfied with workplace design in the activity—based office than individuals who had been located in a private or shared office before relocation. No significant associations regarding satisfaction with office design were found for gender or education.

**3.1.3. Multivariate association between individual factors and change in satisfaction with workplace design after relocation.** The multivariate analysis (Table 4) showed that management position (B = 1.195, p = <0.001) and office type at baseline (B = 1.74, p<0.001) predicted the change in satisfaction with workplace design.

Thus, both univariate and multivariate analyses revealed that position and office type before relocation predicted the change in satisfaction with workplace design. Managers were more satisfied with the new workplace, as were those who had worked in open-plan offices before relocation.

**Table 4. Satisfaction regarding workplace design in the activity based office, compared to before the relocation.** Multivariate linear regression, n = 422.

|  | B (95% CI) | p | Adjusted $R^2$ |
|---|---|---|---|
| **Intercept** | -3.282 (-4.673 to -1.892) | <0.001 |  |
| **Age**: *Years* | -0.015 (-0.033 to 0.004) | 0.122 | 0.209 |
| **Gender** *ref female* | -0.292 (-0.645 to 0.060) | 0.104 |  |
| **Education** *ref not university* | -0.178 (-0.563 to 0.208) | 0.366 |  |
| **Position** *ref employee* | 1.195 (0.551 to 1.838) | **<0.001** |  |
| **Office before relocation** *ref cell office* | 1.741 (1.357 to 2.126) | **<0.001** |  |

Note: The dependent variable is the difference in the index for satisfaction regarding office design, nine months after relocation compared to three months before relocation.

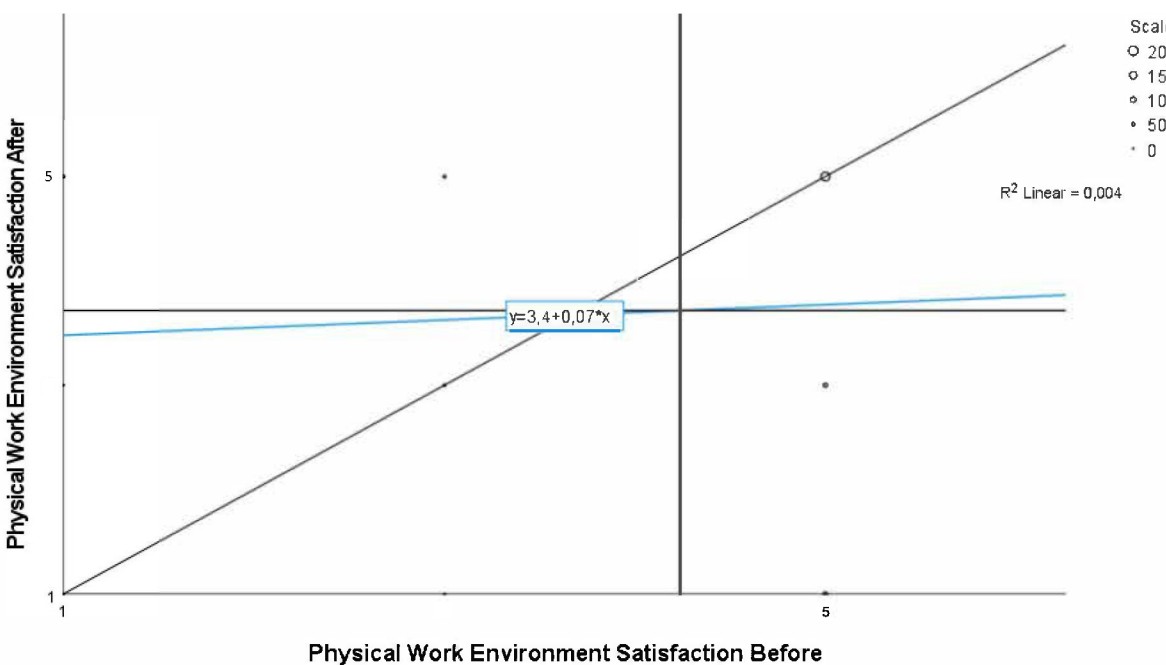

**Fig 2. Physical work environment satisfaction before and after relocation.** Individual results, n = 422. Note: The vertical line shows the group mean at baseline, and the horizontal line shows the group mean at follow-up. Line of identity is included in the diagram, as well as the regression between satisfaction before and after the relocation.

## 3.2 Physical work environment

**3.2.1 Physical work environment satisfaction before and after relocation.** On average, workers were less satisfied with the physical work environment after relocation to activity-based workplaces compared to before (Fig 2). For some workers, the satisfaction increased and for others it decreased, and the association between ratings before and after was weak (Spearman's rank correlation 0.070).

**3.2.2 Univariate association between individual factors and change in satisfaction with the physical work environment after relocation.** Univariate regression analyses are shown in Table 5. Age was negatively associated with the change in satisfaction with the physical work environment (B = -0.021, p = 0.003), meaning that older workers felt more negatively toward the change than younger workers. Univariate analysis also showed that men were more dissatisfied than women with the change in the physical work environment (B = -0.475, p =

**Table 5. Satisfaction regarding physical work environment in the activity-based office, compared to before the move.** Univariate linear regression, n = 422.

|  | B (95% CI) | p | Adjusted $R^2$ |
|---|---|---|---|
| **Age**: *Years* | -0.021 (-0.035 to -0.007) | **0.003** | 0.018 |
| **Gender** *ref female* | -0.475 (-0.737 to -0.212) | **<0.001** | 0.029 |
| **Education** *ref not university* | 0.200 (-0.089 to 0.490) | 0.175 | 0.002 |
| **Position** *ref employee* | 0.825 (0.330 to 1.320) | **0.001** | 0.025 |
| **Office before relocation** *ref cell office* | 1.079 (0.801 to 1.358) | **<0.001** | 0.124 |

Note: Dependent variable is the difference in satisfaction regarding office design, nine months after relocation compared to three months before relocation.

**Table 6. Satisfaction regarding physical work environment in the activity-based office, compared to before the move.** Multivariate linear regression, n = 422.

|  | B (95% CI) | p | Adjusted R$^2$ |
|---|---|---|---|
| **Intercept** | -1.665 (-2.680 to -0.650) | 0.001 | |
| **Age**: *Years* | -0.007 (-0.020 to 0.007) | 0.329 | 0.163 |
| **Gender** *ref female* | 0.056 (-0.226 to 0.338) | 0.696 | |
| **Education** *ref not university* | -0.423 (-0.681 to -0.166) | **0.001** | |
| **Position** *ref employee* | 0.789 (0.319 to 1.259) | **0.001** | |
| **Office before relocation** *ref cell office* | 1.003 (0.722 to 1.284) | **<0.001** | |

Note: Dependent variable is the difference index for satisfaction regarding office design, nine months after relocation compared to three months before relocation.

<0.001). Significant associations were also shown for position (B = 0.825, p = 0.001), i.e. managers felt less negatively toward the change compared to employees. People who had been located in open-plan offices at baseline were less dissatisfied with the change in workplace design after relocation than those who had been in a private or shared office before relocation (B = 1.079, p = <0.001). No significant associations between education and satisfaction with the physical work environment were found in the univariate analysis.

**3.2.3. Multivariate association between individual factors and change in satisfaction with the physical work environment after relocation.** In the multivariate analysis, negative associations for education were shown, meaning that individuals with no university education were more satisfied with the change in the physical environment compared to those with university education (B = -0.423 p = 0.001). Positive associations were found for management position (B = 0.789, p = 0.001), office type at baseline (B = 1.003, p<0.001), and the physical work environment (Table 6).

Both univariate and multivariate analyses revealed that occupational position and office type at baseline were associated with changes in perceived satisfaction with the physical work environment during relocation to activity-based workplaces.

## 3.3 Psychosocial work environment

**3.3.1 Psychosocial work environment satisfaction before and after relocation.** Respondents were, on average, less satisfied with the psychosocial work environment nine months after relocation to activity-based workplaces compared to before (Fig 3). For some, the satisfaction increased and for others it decreased, and the correlation was weak (Spearman's rank correlation 0.192).

**3.3.2 Univariate association between individual factors and change in satisfaction with psychosocial work environment after relocation.** Age was negatively associated with the change in satisfaction regarding psychosocial work environment (B = -0.014, p = 0.038): older employees reported a greater decrease in satisfaction with the psychosocial work environment than younger employees (Table 7). Significant negative associations were also found for gender (B = -0.296, p = 0.024), i.e. that men showed a larger dissatisfaction with the change in psychosocial work environment than women. Position was also associated with changes in satisfaction (B = 0.588, P = 0.017): managers were less dissatisfied compared to employees. Regarding office type at baseline, individuals who were located in open-plan offices at baseline showed on average a smaller decrease in satisfaction with the psychosocial work environment compared to those coming from a private or shared office (B = 0.571 p<0.001). No significant differences in changed satisfaction with the psychosocial work environment were found for education.

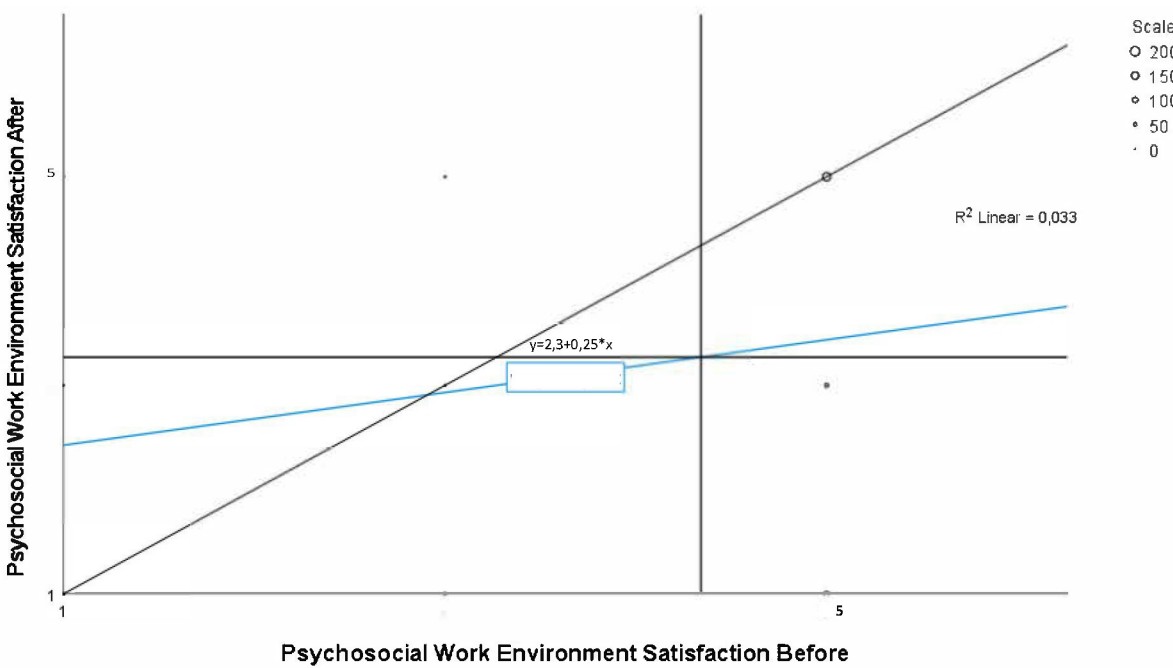

**Fig 3. Psychosocial work environment satisfaction before and after relocation, n = 422.** Note: The vertical line shows the group mean at baseline, whereas the horizontal line shows the group mean at follow-up. Line of identity is included in the diagram, as well as the regression between satisfaction before and after the relocation.

**3.3.3 Multivariate association between individual factors and change in satisfaction with psychosocial work environment after relocation.**  The multivariate analysis for change in satisfaction with the psychosocial work environment (Table 8) showed negative associations for education (B = -0.309, p = 0.022) and positive associations for management position (B = 0.682, p = 0.006) and office type at baseline (B = 0.513, p = 0.001).

Thus, both univariate and multivariate analyses revealed that, position, and office type at baseline were associated with changes in perceived satisfaction with the psychosocial work environment during relocation to activity-based workplaces.

## 4. Discussion

This prospective study contributes to a better understanding of the different individual factors influencing changes in satisfaction with the work environment during relocation to activity-based workplaces, and the degree of these changes.

**Table 7.  Satisfaction regarding psychosocial work environment in the activity-based office, compared to before the move.**  Univariate linear regression, n = 422.

|  | B (95% CI) | p | Adjusted $R^2$ |
|---|---|---|---|
| **Age**: *Years* | -0.014 (-0.028 to -0.001 | **0.038** | 0.010 |
| **Gender** *ref female* | -0.296 (-0.553 to -0.039) | **0.024** | 0.012 |
| **Education** *ref not university* | -0.051 (-0.333 to 0.231) | 0.722 | -0.002 |
| **Position** *ref employee* | 0.588 (0.105 to 1.071) | **0.017** | 0.013 |
| **Office before relocation** *ref cell office* | 0.571 (0.287 to 0.855) | **<0.001** | 0.035 |

Note: The dependent variable is the difference in satisfaction regarding office design, nine months after relocation compared to three months before relocation.

**Table 8. Satisfaction regarding psychosocial work environment in the activity-based office, compared to before the move.** Multivariate linear regression, n = 422.

|  | B (95% CI) | p | Adjusted $R^2$ |
|---|---|---|---|
| **Intercept** | -1.173 (-2.218 to -0.127) | 0.028 | |
| **Age**: *Years* | -0.009 (-0.023 to 0.005) | 0.208 | 0.060 |
| **Gender** *ref female* | -0.156 (-0.446 to 0.134) | 0.292 | |
| **Education** *ref not university* | -0.309 (-0.574 to -0.044) | **0.022** | |
| **Position** *ref employee* | 0.682 (0.198 to 1.166) | **0.006** | |
| **Office before relocation** *ref cell office* | 0.513 (0.224 to 0.803) | **0.001** | |

Note: The dependent variable is the difference index for satisfaction regarding office design, nine months after relocation compared to three months before relocation.

We found that satisfaction with the workplace design, the physical work environment, and the psychosocial work environment decreased after relocation to activity-based workplaces. The finding of decreased work environment satisfaction when moving to activity-based workplaces is in line with previous research [7, 8, 11, 12]. However, the present study showed that the change in satisfaction varied depending on different individual factors, and also that it differed considerably between workers. For some, satisfaction increased and for others it decreased after relocation, and the responses before and after the move correlated weakly. Results from both univariate and multivariate models indicated that occupational position and office type before relocation predict the change in work environment satisfaction during relocation to activity-based workplaces. Specifically, managers and those working in an open-plan office before relocation reported a smaller decline in satisfaction after relocation, compared to those coming from private or shared offices. This finding was consistent for all three outcomes, i.e. satisfaction with workplace design, physical work environment, and psychosocial work environment, with smaller effect sizes in models predicting changes in psychosocial satisfaction. Also, education was in the univariate analysis a significant predictor of the change in satisfaction with the physical and psychosocial work environment, but not of the change in satisfaction with office design. We also found significant effects of age and gender on changes in satisfaction, with younger individuals and women being less affected by the relocation to activity-based workplaces.

Occupational position might relate to work tasks and different work tasks might require different conditions. In a systematic review, Engelen et al. [11] found that communication, social interaction, and work time control are related to work satisfaction in activity-based workplaces. It could be that it is easier to lead and distribute work when managing others in the same environment without separate enclosed rooms, because it can facilitate better overview and easier interaction. However, we did not ask about work tasks or control for specific occupational roles, which may be a limitation of this study.

Those who worked in an open-plan office before relocation had a smaller decrease in satisfaction with the work environment when relocated to an activity-based workplace than those who worked in a private office at baseline. One explanation for this could be that open-plan offices are more similar to activity-based workplaces, and thus moving to the new office will not imply the same extent of change. Another explanation could be that individuals who were previously located in an open-plan office are more used to distractions during work. Hovath et al. [26] found individual differences in distraction at work during performance of an auditory task. Another explanation for the variation in satisfaction can be found in a study

reporting that private offices are associated with greater satisfaction in terms of privacy and the ability to concentrate, while conversely they hinder communication [37]. Jahncke et al. [37] found that in activity-based offices, task performance was better in quiet work zones than in active zones. Perhaps not surprisingly, previous studies have shown that employees communicate more in open-plan offices, or interact differently, which might contribute to higher satisfaction among workers who need or simply like communication [12, 38, 39]. However, in contrast, a field study showed that face-to-face interaction decreased in more open office spaces, while digital interaction increased [40]. This is not what companies are aiming for with open-plan or activity-based offices, and highlights the need for more research into behaviors in activity-based workplaces.

Respondents with no university education showed significantly less change in satisfaction with the physical and psychosocial work environment after relocation than before, compared to respondents with university education. However, the beta coefficients for education changed in the multivariate analysis compared to the univariate, which may be a sign of confounding. In the survey, we asked for educational level using four categories, which we merged into two in the analysis: *university* or *no university*, in order to get comparable group sizes. At this particular agency, employees' educational background was quite homogenous. If the sample had been larger and the educational requirements for the tasks more diverse, we could have compared education in more categories. In future research, it would be interesting to explore how education predicts work satisfaction in activity-based workplaces, since education appears to determine job satisfaction in general [30] We also found age differences in this study, in line with previous studies on work satisfaction [23, 41].Many studies have addressed gender differences in job satisfaction in general. As one example, Mason [42] found that men and women in management positions did not differ. Another example is a study by MottazI [43] that identifies gender differences in work satisfaction, and explains this in terms of differences in values and expectations. A number of factors moderate satisfaction for the individual, including need for privacy, complexity of tasks, and occurrence of misfit in the office [3]. Wohlers et al. [32] found that it is important that office workers use activity-based work environments according to task requirements, and that appropriate use of the activity-based work environment correlates negatively with workers' need for routine seeking. Also, contextual factors might contribute to satisfaction in activity-based workplaces. For instance, De Been and Beijer [4] found that physical work satisfaction and productivity, for both men and women, were associated with indoor climate. Previous studies have also demonstrated that women appear to be more critical toward the indoor climate as they are more sensitive to both hot and cold room temperatures [44].

In the present study, we found that men were more negative toward the changes following from relocation to activity-based workplaces than women in a univariate analysis, while we did not find a significant difference between genders in the multivariate analysis. Previous studies have found other gender differences in exposure to job stressors, i.e. women are also more vulnerable to stressors, with factors such as control, marital status, and income having an effect [45]. In general, it is not only the design of the workplace that matters, but factors such as career opportunities and how long a person has been in a workplace, in combination with getting older, affect job satisfaction [23]. It is therefore likely that activity-based workplaces alone cannot account for work satisfaction. Rather, contextual factors need to be taken into account, such as climate and personal attitudes. A recent Norwegian study reported effects of personality, such as gender and health status, including that open and shared workspace designs increase the risk of disability retirement among office workers [46].

### 4.1 Methodological discussion

A strength of the present study is that all employees at two large agencies had the opportunity to answer the questionnaire at baseline and follow-up, providing a large number of respondents (n = 422). However, there are also limitations. When asking people to self-rate satisfaction, there is always a risk that one's satisfaction with life in general will influence the ratings given in a questionnaire. This risk at the individual level is reduced when many individuals answer the same question, since the influence of overall life satisfaction may be positive for some and negative for others. Another limitation of the study is that there may be individual characteristics in addition to age, gender, education, position, and office type before relocation that may potentially influence satisfaction. However, we did not have access to any further information on the participants. Personality would also have been interesting to study, but it would have required a different approach than a web survey. The present prospective study used a follow-up time of nine months, which may be insufficient to capture long-term changes in satisfaction. It is possible that initial effects fade as people get accustomed to the new working situation. For this reason, a long-term follow-up is desirable in surveys of workplace satisfaction, including long-term results regarding new ways of interacting and communicating in activity-based workplaces. The measurements in the present study were performed one year apart. To our knowledge, there were no other major changes at the agency during this period that might have contributed to the decline in satisfaction with the work environment.

We chose to include only those who answered the questionnaire both at baseline and follow-up in order to implement a repeated measures design. Since there were some respondents who were not working at the agency on both these occasions, or for other reasons did not answer both questionnaires, there was a loss of respondents between baseline and follow-up (Table 2). We recruited participants via email, which may have reduced the willingness to answer [47]. Dropouts were similar to the study sample in gender and position, but differed slightly in age, education, and office type at baseline. This might have influenced the generalizability of our findings, although we expect any eventual bias to be small.

In future research, we encourage further exploration of individual factors that may contribute to work environment satisfaction with respect to specific work tasks, and the extent to which work-related values contribute to satisfaction in different groups of individuals.

## 5. Conclusion

On average, satisfaction decreased after relocation to activity-based workplaces, but for some individuals, satisfaction in fact increased. Occupational position and office type at baseline were determinants of the change in work satisfaction: managers showed a smaller decrease in satisfaction than employees, and those who worked in open-plan offices before relocation were less dissatisfied with the relocation than those coming from private offices. Participants with university education experienced a larger negative change in satisfaction with the physical and psychosocial work environment than those who had not attended university. Thus, we can confirm the results of previous studies showing that work environment satisfaction decreases when relocating to activity-based workplaces, but also that the degree of change varies considerably between individuals.

## 6. Practical implications

When planning relocation to activity-based workplaces, the characteristics of the group being relocated should be taken into consideration in order to provide the best possible conditions for a satisfying working environment. For those planning to introduce activity-based workplaces, the present study offers evidence that work position, office type before relocation, and

level of education matter. Individuals located in open-plan offices before a relocation are more likely to be satisfied with activity-based workplaces than those relocating from private or shared offices.

## Supporting information

**S1 File.**
(XLSX)

## Author Contributions

**Conceptualization:** Katarina Wijk, Eva L. Bergsten, David M. Hallman.

**Data curation:** Katarina Wijk, Eva L. Bergsten, Svend Erik Mathiassen, David M. Hallman.

**Formal analysis:** Katarina Wijk, David M. Hallman.

**Funding acquisition:** Katarina Wijk, Eva L. Bergsten, David M. Hallman.

**Methodology:** Katarina Wijk, Eva L. Bergsten, Svend Erik Mathiassen, David M. Hallman.

**Validation:** Katarina Wijk, Eva L. Bergsten, Svend Erik Mathiassen, David M. Hallman.

**Visualization:** Katarina Wijk.

**Writing – original draft:** Katarina Wijk.

**Writing – review & editing:** Eva L. Bergsten, Svend Erik Mathiassen, David M. Hallman.

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
