## [Decision Letter · Decision Letter 0]

13 Jun 2022

PONE-D-21-33071Individual determinants of satisfaction with the work environment after relocation to activity based workplaces: a prospective studyPLOS ONE

Dear Dr. Wijk,

Thank you for submitting your manuscript to PLOS ONE. After careful consideration, we feel that it has merit but does not fully meet PLOS ONE’s publication criteria as it currently stands. Therefore, we invite you to submit a revised version of the manuscript that addresses the points raised during the review process.

 This manuscript reports on a study of factors determining satisfaction before and after relocation of public office workers to an activity-based workplace. There is however no given definition of or criteria for «private or shared office» or «open plan office» at baseline. A revised manuscript should follow through the recommendations from both reviewers and should present the data showing the number of managers and employees, university degree and elementary training, age groups occupying the various office types at baseline, and their respective satisfaction reports. 

We look forward to receiving your revised manuscript.

Kind regards,

Denis Alves Coelho, PhD

Academic Editor

PLOS ONE

Journal Requirements:

Additional Editor Comments (if provided):

Please ponder and act upon the reviewer recommendations provided.

Reviewers' comments:

Reviewer's Responses to Questions

**Comments to the Author**

1. Is the manuscript technically sound, and do the data support the conclusions?

Reviewer #1: Yes

Reviewer #2: Partly

2. Has the statistical analysis been performed appropriately and rigorously? 

Reviewer #1: Yes

Reviewer #2: Yes

3. Have the authors made all data underlying the findings in their manuscript fully available?

Reviewer #1: Yes

Reviewer #2: No

4. Is the manuscript presented in an intelligible fashion and written in standard English?

Reviewer #1: Yes

Reviewer #2: Yes

5. Review Comments to the Author

Reviewer #1: PONE-D-21-33071

This paper describes a study of factors determining satisfaction before and after relocation of public office workers to an activity-based workplace (ABW).

ABW has become the buzzword of architects/consultants and employers with promise of several advantages compared to individual cell offices, one of which is lower costs of workplace space. Hence, knowledge of consequences for satisfaction, motivation, function, and health is needed, and the potential practical impact of the present study is high.

A major problem of studying ABW is the definition of the concept. The present paper describes ABW on p 3: “In activity based workplaces employees share a non-assigned area designed for flexibility and different types of settings e.g. open meeting spaces, closed rooms, and silent spaces allowing different types of activities e.g. interaction with colleagues or silent concentration….” This can mean anything from landscapes with labelled zones to combinations of cell offices, meeting rooms, and remote work. Hence, important dimensions that may define how employees perceive and perform work are not defined: (a) number of employees in the room space, (b) area per employee, (c) fixed versus non-fixed workplace (shared seating), (d) number of workstations relative to number of employees (i.e., availability of preferred workstation, availability of workstation close to team members), (e) level of remote work.

The present paper does not define or describe the ABW studied, hence the external validity is limited. Did all employees have shared seating/clen desk? Did managers have the same work situation as all others in the ABW? Furthermore, the work tasks performed are not described. Time spent on solitary work, time spent in telephone conversations with clients or suppliers, and time spent in meetings define the needs of the employees. There is no description or data of these parameters and description of work tasks. There is no definition of or criteria for “private or shared office” or “open plan office” at baseline.

The authors conclude that “Results from both univariate and multivariate models indicate that occupational position and office type before relocation predict the change in work environment satisfaction during relocation to activity based workplaces.” This is based on standard epidemiological analyses and there is no reason to doubt these findings. However, these analyses must be supplemented by specific information of the participants’ subgroups. For instance, the number of managers (or high-level of education) were working in individual offices at baseline. The paper should present the data showing the number of managers and employees, university degree and elementary etc training, age groups occupying the various office types at baseline, and their respective satisfaction reports. Data of baseline versus follow-up means (or medians) of all these subgroups would strengthen the quality and usability of information of the paper (even if some groups would be too small for testing statistical significance).

According to Figures 2 and 3, the mean levels of physical and psychosocial satisfaction ranged from 2.2 to 2.7. These numbers are very low – are they correct?

The Discussion contains references to studies of job satisfaction and of gender effects. The authors should either delete references that are not related to office concepts or do a comprehensive summary of the hundreds of studies in these fields.

The authors maintain that “Previous studies have shown that there is more communication in open offices, which might contribute to a better satisfaction among workers who need or like communication (Banbury & Berry, 2005; van der Voordt, 2004).” This is controversial and the recent study by Bernstein and Turban (Philosophical Transactions B, 2018) who showed with objective recording methods that face-to-face interactions decreased by ca 70%, should be included an discussed.

The authors stated that “In future research, we encourage to further explore individual factors that can contribute to work satisfaction with respect to specific work tasks, and to explore the extent to which work related values contribute to work satisfaction in different groups of individuals.” A study by Nielsen and co-workers (Scand J Work Environ Health, 2021) reported effects of personality type and office type on disability retirement. The pronounced effects reported in that study illustrate the necessity to define criteria for defining a workplace as open-plan office versus ABW.

Reviewer #2: 1) Introduction, 1.1: Please review if some of the past findings referred to might in fact overlap – does each publication need their own sentence? Also, the heading “1.1 satisfaction in activity based workplaces” should be considered changed as several of the studies referred to focuses on physical work environment, work engagement, health, psychosocial working conditions – aspects not equivalent to “satisfaction” per se.

2) Introduction 1.2 second section page 5: It seems unclear why activity based work environments are discussed in relation to associations between “time on a particular job” and “satisfaction”. Is it to highlight that activity based work environments potentially might moderate impacts of time on a particular job on satisfaction?

3) 2.1 Please specify the statistical analysis for determining differences between responders and non-responders. And you might perhaps consider including the analysis in appendix.

4) The sentence “The questionnaire was constructed and administered using Webropol (..)” is written twice.

5) Please specify the purpose of the univariate analyses. It seems that you - regardless of significance/non-significance – adjust for all other variables in the multivariate models anyway.

6) Some consideration is given to why “education” suddenly becomes significant in the multivariate analyses.

7) 3.3.3: It is stated below table 8 that - univariate analyses reveal that education is associated with the outcome - which does not seem to be the case given table 7.

8) How strongly do the three outcome measures correlate? Satisfaction with “the physical” environment and satisfaction with “the psychosocial” environment appears to be global general measures while satisfaction with workplace design taps into specific aspects of the physical and psychosocial environment. Thus, there seems to be (conceptually) a distinctive overlap between the outcome measures and it is not clear.

6. PLOS authors have the option to publish the peer review history of their article (what does this mean?). If published, this will include your full peer review and any attached files.

Reviewer #1: No

Reviewer #2: No

---

## [Author Response · Author response to Decision Letter 0]

23 Aug 2022

Response to reviewers 

Individual determinants of satisfaction with the work environment after relocation to activity based workplaces: a prospective study 

Thanks for all comments, below you see how we have improved the manuscript in accordance with your comments. 

Comments to each suggestion:

We have ensured that the manuscript meets the Plos Ones style requirement

The references has been changed to Vancocer style 

Level 1 headings are bold type 18pt font, and level 2 heading bold type 16pt, level 3 14pt

Cited Figures written as Fig

Reviewer 1

Comment: A major problem of studying ABW is the definition of the concept. The present paper describes ABW on p 3: “In activity based workplaces employees share a non-assigned area designed for flexibility and different types of settings e.g. open meeting spaces, closed rooms, and silent spaces allowing different types of activities e.g. interaction with colleagues or silent concentration….” This can mean anything from landscapes with labelled zones to combinations of cell offices, meeting rooms, and remote work. Hence, important dimensions that may define how employees perceive and perform work are not defined: (a) number of employees in the room space, (b) area per employee, (c) fixed versus non-fixed workplace (shared seating), (d) number of workstations relative to number of employees (i.e., availability of preferred workstation, availability of workstation close to team members), (e) level of remote work.

The present paper does not define or describe the ABW studied, hence the external validity is limited. Did all employees have shared seating/clen desk? Did managers have the same work situation as all others in the ABW? Furthermore, the work tasks performed are not described. Time spent on solitary work, time spent in telephone conversations with clients or suppliers, and time spent in meetings define the needs of the employees. There is no description or data of these parameters and description of work tasks. There is no definition of or criteria for “private or shared office” or “open plan office” at baseline.

Response: This has been clarified in 2.1: Employees at the agency that did not relocate or had received customized work places were also excluded, meaning that all participants regardless of position or work tasks in the study work in activity based workplaces after relocation. The duties at the agency were mainly administrative, although the profession varied. The non-assigned area designed for flexibility and different types of activity was divided in labelled zones i.e. quiet zones and meeting rooms. Before the relocation it was ensured by a designated project group that the activity based workplace will be dimensioned to be enough for everyone. During the working day one are supposed to shift zone as tasks varies, meaning that the needs of the employee govern where one work. Before relocation a set of preparation activities i.e. education, information was conducted in order to ensure that everyone know office rules etc. 

Further a defintion has been added: Office type before relocation was divided in two categories, in both groups you had a dedicated seat. Private or shared office mean that one to four persons are seated in one cell office with a door that can be closed, while open plan office are larger and shared by many. 

We also add a referens to previous paper where this is more extensively described. Bergsten, E.L.; Wijk, K.; Hallman, D.M. Relocation to Activity-Based Workplaces (ABW)—Importance of the Implementation Process. Int. J. Environ. Res. Public Health 2021, 18, 11456. https://doi.org/10.3390/ ijerph182111456

Comment: The authors conclude that “Results from both univariate and multivariate models indicate that occupational position and office type before relocation predict the change in work environment satisfaction during relocation to activity based workplaces.” This is based on standard epidemiological analyses and there is no reason to doubt these findings. However, these analyses must be supplemented by specific information of the participants’ subgroups. For instance, the number of managers (or high-level of education) were working in individual offices at baseline. The paper should present the data showing the number of managers and employees, university degree and elementary etc training, age groups occupying the various office types at baseline, and their respective satisfaction reports. Data of baseline versus follow-up means (or medians) of all these subgroups would strengthen the quality and usability of information of the paper (even if some groups would be too small for testing statistical Response: A new table (table 2) has been added with information on subgroups showing the number of managers and employees, university degree and elementary etc training, age groups occupying the various office types at baseline, and their respective satisfaction reports, showing mean, percent, number, SD . This table has been added in the result section 3. Previous table 1 has been changed and taken away from method section. 

Comment: According to Figures 2 and 3, the mean levels of physical and psychosocial satisfaction ranged from 2.2 - to 2.7. These numbers are very low – are they correct?

Respons: In figure 2 and 3 we used three categories to illustrate the result. From reviewers feedback we relize that it was not clear, thanks. We have now changed the figures 2 and figure 3 so that the x and y contain the scale 1-5 instead of categories (mean value). 

Comment: The Discussion contains references to studies of job satisfaction and of gender effects. The authors should either delete references that are not related to office concepts or do a comprehensive summary of the hundreds of studies in these fields. 

Response: We have clarified in the discussion that the references on previous studies on gender and job satisfaction only are examples. We like to kep the references in order to discus our results in relation to prevous studies on gender and job satisfaction but do not se the neccesity of summarizing hundreds of studies. 

Comment: The authors maintain that “Previous studies have shown that there is more communication in open offices, which might contribute to a better satisfaction among workers who need or like communication (Banbury & Berry, 2005; van der Voordt, 2004).” This is controversial and the recent study by Bernstein and Turban (Philosophical Transactions B, 2018) who showed with objective recording methods that face-to-face interactions decreased by ca 70%, should be included an discussed.

Response: We have added the reference in the discussion: Bernstein, Ethan, and Stephen Turban. "The Impact of the 'Open' Workspace on Human Collaboration." Art. 239. Philosophical Transactions of the Royal Society B, Biological Sciences 373, no. 1753 (August 19, 2018).

Comment: The authors stated that “In future research, we encourage to further explore individual factors that can contribute to work satisfaction with respect to specific work tasks, and to explore the extent to which work related values contribute to work satisfaction in different groups of individuals.” A study by Nielsen and co-workers (Scand J Work Environ Health, 2021) reported effects of personality type and office type on disability retirement. The pronounced effects reported in that study illustrate the necessity to define criteria for defining a workplace as open-plan office versus ABW.

Response: This reference has been added and commented in the discussion

Reviewer 2

Comment: Introduction, 1.1, page 4, first section: Please review if some of the past findings referred to might in fact overlap. does each publication need their own sentence

Response: We have rewritten the introduction some and clarified that from previous studies we do not know if individual factors i.e. age, gender, education, working position and office type at baseline predict work satisfaction in activity based workplaces.

Comment: Also, the heading “1.1 satisfaction in activity based workplaces” should be considered changed as several of the studies referred to focuses on physical work environment, work engagement, health, psychosocial working conditions – aspects not equivalent to “satisfaction” per se. 

Response: Title 1:1 has been changed

Comment: Please also consider offering a definition of “satisfaction”, alternatively an operational definition. 

Response: A definition are added in 2.3: We used work environment satisfaction, focusing satisfaction with the physical and psychosocial work environment, as a measure of the overall perception of the work environment.

Comment: Introduction 1.2 second section page 5: It seems unclear why activity based work environments are discussed in relation to associations between “time on a particular job” and “satisfaction”. Is it to highlight that activity based work environments potentially might moderate impacts of time on a particular job on satisfaction?

Response: Thanks, this section in introduction 1.2 has been rewritten. What we want to describe is that “workers perception of work environment may change during ones working life”, inferring that age might contribute to perception of working in activity based work places. 

Comment: 2.1 Please specify the statistical analysis for determining differences between responders and non-responders. And you might perhaps consider including the analysis in appendix.

Response: Thank you. We have rewritten and added a new table in the result section table 2 section. The table contain descriptive data; mean, number, percent and SD. 

Since the response rate was high at baseline, the interesting thing to know would be if there are differences between respondents participating at both occasions, compared to those only answering at baseline, since we compare only those who had answered both times.. information in section 2.1 that the analysis was descriptive and that there were no major differences in terms of age, gender, education, position or office type among respondents that received questionnaires and did not answer after relocation, compared to those who did answer both 

Comment: The sentence “The questionnaire was constructed and administered using Webropol (..)” is written twice. 

Response: Thanks, this has been changed 

Comment: It is stated several places that certain aspects are associated with “less decreased satisfaction” but doesn’t the results actually show an increase in satisfaction (positive Beta values)? Also related to this, please check grammar and consider wording. 

Response: The results show decreases in satisfaction as described in 3.1, 3.2, 3.3. Nevertheless some persons in the group show less decreases than others, for instance, as described in 3.3.2, women show less decrease in satisfaction compared to men. This does not mean that satisfaction increase for woman but that the decreases were more pronounced for men. Proofreading has been conducted by a professional proofreader; you can see most of the changes in the manuscript. 

Comment: Please specify the purpose of the univariate analyses. It seems that you - regardless of significance/non-significance – adjust for all other variables in the multivariate models anyway. 

Response: Our aim was to determine the extent to which factors age, gender, education working position and office type at baseline predict satisfaction. Univariate analyses make it possible to specify the relationship between two variables, for instance age and satisfaction. We wished, however, to also see the composite relationship, using several variables as determinants. Those two analyses are complementary since univariate analyses do not allow for confounding factors to be taken into account

Comment: Some consideration is given to why “education” suddenly becomes significant in the multivariate analyses compared the univariate analyses but it doesn’t seem to be sufficiently covered. 

3.3.3: It is stated below table 8 that - univariate analyses reveal that education is associated with the outcome - which does not seem to be the case given table 7.

Response: This is, as described above, an example of how univariate and multivariate analysis are complementary and give us additional information. The relationship between education and satisfaction are not significant, but when several variables are analyzed together, education seems to predict satisfaction. In the discussion, we have addressed this. We do not have information on why education predicts satisfaction. 

Comment: How strongly do the three outcome measures correlate? Satisfaction with “the physical” environment and satisfaction with “the psychosocial” environment appears to be global general measures while satisfaction with workplace design taps into specific aspects of the physical and psychosocial environment. Thus, there seems to be (conceptually) a distinctive overlap between the outcome measures and it is not clear, in particular, what the analyses with the two global measures add.

Response: This is an interesting question. We believe that the measures contribute to valuable knowledge on self-reported perceptions about overall physical and psychosocial work environment while the six questions about workplace design ask more specifically about opportunity to reach colleagues, physical distance between colleagues, size of workplace, design and technical tools. All results show the same tendency, i.e. decreases in satisfaction after relocation, but we have not measured correlations between the variables in this paper. In another paper it could be interesting to measure correlation between different outcome measures, especially if the aim is to further validate measurements. 

Comment: There is however no given definition of or criteria for «private or shared office» or «open plan office» at baseline. 

Respons: A definition has been added in 2.3. In a ‘private or shared office’ one to four persons share one cell office with a door that can be closed, while an ‘open plan office’ is larger and shared by many. In the present study, workers had a personal working place in both office types.

Comment: A revised manuscript should should present the data showing the number of managers and employees, university degree and elementary training, age groups occupying the various office types at baseline,

Response:The number in each group working in private/shared office vs. open plan office before relocation has been added in a new table (table 2) in section 3.

---

## [Decision Letter · Decision Letter 1]

22 Nov 2022

PONE-D-21-33071R1Individual determinants of satisfaction with the work environment after relocation to activity based workplaces: a prospective studyPLOS ONE

Dear Dr. Wijk,

Thank you for submitting your manuscript to PLOS ONE. After careful consideration, we feel that it has merit but does not fully meet PLOS ONE’s publication criteria as it currently stands. Therefore, we invite you to submit a revised version of the manuscript that addresses the points raised during the review process.

Both reviewers recommend  proceeding with the manuscript for publication. however, the reviewer that has read your manuscript in the revised stage for the first time has raised some minor questions, that we would be happy to see clarified in the second revision of this manuscript.

We look forward to receiving your revised manuscript.

Kind regards,

Denis Alves Coelho, PhD

Academic Editor

PLOS ONE

Journal Requirements:

Reviewers' comments:

Reviewer's Responses to Questions

**Comments to the Author**

1. If the authors have adequately addressed your comments raised in a previous round of review and you feel that this manuscript is now acceptable for publication, you may indicate that here to bypass the “Comments to the Author” section, enter your conflict of interest statement in the “Confidential to Editor” section, and submit your "Accept" recommendation.

Reviewer #3: (No Response)

Reviewer #4: All comments have been addressed

2. Is the manuscript technically sound, and do the data support the conclusions?

Reviewer #3: Partly

Reviewer #4: Yes

3. Has the statistical analysis been performed appropriately and rigorously? 

Reviewer #3: Yes

Reviewer #4: Yes

4. Have the authors made all data underlying the findings in their manuscript fully available?

Reviewer #3: No

Reviewer #4: Yes

5. Is the manuscript presented in an intelligible fashion and written in standard English?

Reviewer #3: Yes

Reviewer #4: No

6. Review Comments to the Author

Reviewer #3: Thank you for the opportunity to review this revised draft of the manuscript which addresses an interesting and relevant topic given the rapid growth in ABW environments. The authors have in most cases addressed the earlier reviewer comments on the initial submission but in my view several minor issues remain to be addressed in the manuscript before publication.

1. Introduction: What process was followed to select the (five) specific independent individual worker characteristics/variables (i.e. baseline age, gender, education, working position and office type) to the exclusion of other potential individual characteristics such as personality type which may affect satisfaction with transition to ABW?

2. Methods: Was there any attempt in the survey to describe or define for respondents what they should understand by the terms Physical work environment and psychosocial work environment in answering the respective single item questions (“Regarding your work in general, how satisfied are your regarding the physical work environment?”, and "Regarding your work in general, how satisfied are your regarding the psychosocial environment"). These terms may mean different terms to different individuals, particular the psychosocial work environment. In contrast satisfaction with workplace design was limited specifically to 6 questions.

3. Results: were mostly clear. However in Table 2 there is an error in percentage of elementary school education in the total sample (32.9.4). Additionally, Page 19 (line 1-2) incorrectly reports that".. both univariate and multivariate analyses reveal that education, position......associated with changes in perceived satisfaction with the psychosocial work environment during relocation to activity based workplaces". This finding is not the case for univariate analysis (see Table 3 and line 2-3 page 18).

4. Discussion- Methodological limitation should acknowledge that some individual characteristics such as personality type were not included in the study as independent factors potentially influencing worker satisfaction.

6. Practical implications: The statement "A workplace with, for example mainly managers may find it easier to gain

acceptance for activity based workplaces, compared to a group of workers"; does not accord with the findings of the research or indeed the conclusion above on page 24 "employees showed less decrease in satisfaction than managers".

Reviewer #4: Although I recommended accepting the manuscript, however, the manuscript requires proofread by native English person. There are many simple structure errors and grammatical mistakes.

7. PLOS authors have the option to publish the peer review history of their article (what does this mean?). If published, this will include your full peer review and any attached files.

Reviewer #3: No

Reviewer #4: No

---

## [Author Response · Author response to Decision Letter 1]

5 Jan 2023

Comments to each point raised:

Reviewer #3

1. Introduction: What process was followed to select the (five) specific independent individual worker characteristics/variables (i.e. baseline age, gender, education, working position and office type) to the exclusion of other potential individual characteristics such as personality type which may affect satisfaction with transition to ABW?

Response: In section 2:1, we refer to previous studies (references 3, 19-32) indicating that job satisfaction in general may be related to age, gender, education, office type and seniority. On page 5, we argue that “Given that a growing body of literature emphasizes the influence of activity-based workplaces on work satisfaction in general, and the importance of individual factors in particular, we find it useful to further examine the extent to which different individual factors have an impact on work satisfaction in activity-based workplaces”.

We considered controlling for job title, but chose not to since tasks vary greatly within the professional categories; for example, administrators may carry out a wide variety of individual administrative tasks, including various interactive tasks. Personality would also have been interesting to study, but this would have required a different approach than a web survey. We have added this limitation to the methodological discussion in section 4:1 

2. Methods: Was there any attempt in the survey to describe or define for respondents what they should understand by the terms Physical work environment and psychosocial work environment in answering the respective single item questions (“Regarding your work in general, how satisfied are your regarding the physical work environment?”, and "Regarding your work in general, how satisfied are your regarding the psychosocial environment"). These terms may mean different terms to different individuals, particular the psychosocial work environment. In contrast satisfaction with workplace design was limited specifically to 6 questions.

Response: As mentioned in section 2:3, we asked for perceived overall satisfaction with physical and psychosocial work environment and did not define these concepts, which has been added as a limitation in the discussion. In section 2:3, we have also clarified that we asked for perceived overall work environment satisfaction. 

3. Results: were mostly clear. However in Table 2 there is an error in percentage of elementary school education in the total sample (32.9.4). Additionally, Page 19 (line 1-2) incorrectly reports that".. both univariate and multivariate analyses reveal that education, position......associated with changes in perceived satisfaction with the psychosocial work environment during relocation to activity based workplaces". This finding is not the case for univariate analysis (see Table 3 and line 2-3 page 18).

Response: Thank you, this mistake has been corrected in the manuscript in Table 2 and on page 19. 

4. Discussion: Methodological limitation should acknowledge that some individual characteristics such as personality type were not included in the study as independent factors potentially influencing worker satisfaction.

Response: Thank you for this comment. We agree that other factors, such as personality type, may be of importance, and this has been added to the discussion in section 4:1. 

5. Practical implications: The statement "A workplace with, for example mainly managers may find it easier to gain acceptance for activity based workplaces, compared to a group of workers"; does not accord with the findings of the research or indeed the conclusion above on page 24 "employees showed less decrease in satisfaction than managers".

Response: Thank you for this important comment. We have revised the manuscript by removing the sentence from the manuscript in order to make sure there will be no misunderstandings. What we intended to say is that employees were less satisfied compared to managers and that managers generally rated questions about satisfaction higher than employees did. 

Reviewer #4: 

1.Although I recommended accepting the manuscript, however, the manuscript requires proofread by native English person. There are many simple structure errors and grammatical mistakes.

Response: A professional proofreader has carefully corrected grammatical and structural errors, see track changes in manuscript.

Have the authors made all data underlying the findings in their manuscript fully available?

Reviewer #3: No; Reviewer #4: Yes

Response: The data contain sensitive information. Due to the ethical restrictions imposed by the Swedish Ethical Review Authority (at the time of the study: Regional Ethical Review Board in Uppsala), original data more detailed than the attached minimal dataset are available only upon request. Reasonable requests for data may be sent to the following email address, with due reference to the name of the project: registrator@hig.se.

---

## [Decision Letter · Decision Letter 2]

16 Jan 2023

PONE-D-21-33071R2

Individual determinants of satisfaction with the work environment after relocation to activity-based workplaces: a prospective study

PLOS ONE

Dear Dr. Wijk,

Thank you for submitting your manuscript to PLOS ONE. After careful consideration, we feel that it has merit but does not fully meet PLOS ONE’s publication criteria as it currently stands. Therefore, we invite you to submit a revised version of the manuscript that addresses the points raised during the review process.

The authors need careful attention to the presentation of the Tables 3-8 and make sure consistencies in the results of the statistical analysis and statements in the body of the manuscript. The authors may need to seek advice of experts of statistical analysis to make sure the interpretation of the results and academic presentation of the results of statistical analysis.

We look forward to receiving your revised manuscript.

Kind regards,

Keiko Nakamura

Academic Editor

PLOS ONE

Journal Requirements:

Reviewers' comments:

Reviewer's Responses to Questions

**Comments to the Author**

1. If the authors have adequately addressed your comments raised in a previous round of review and you feel that this manuscript is now acceptable for publication, you may indicate that here to bypass the “Comments to the Author” section, enter your conflict of interest statement in the “Confidential to Editor” section, and submit your "Accept" recommendation.

Reviewer #3: (No Response)

Reviewer #4: All comments have been addressed

2. Is the manuscript technically sound, and do the data support the conclusions?

Reviewer #3: Yes

Reviewer #4: Partly

3. Has the statistical analysis been performed appropriately and rigorously? 

Reviewer #3: Yes

Reviewer #4: Yes

4. Have the authors made all data underlying the findings in their manuscript fully available?

Reviewer #3: Yes

Reviewer #4: No

5. Is the manuscript presented in an intelligible fashion and written in standard English?

Reviewer #3: Yes

Reviewer #4: Yes

6. Review Comments to the Author

Reviewer #3: I am satisfied that the authors' response letter has addressed all the issues I raised in my last review, and are reflected in the second revised manuscript with the exception of one issue. I previously noted that on page 18 it is stated that no significant differences in changed satisfaction with the psychosocial work environment were found for education. However on page 19 the authors conclude in a statement that both univariate and multivariate analyses revealed that education...was associated with changes in perceived satisfaction with the psychosocial work environment during relocation to activity-based workplaces. This statement may be correct for all other variables but not education. Repetition of this error makes me question the proof reading of the manuscript by the authors given they indicated in their response that the error had been corrected. However, if this issue is addressed I am satisfied that the manuscript is ready for publication.

Reviewer #4: Thank you for addressing most of the comments. However, Table 3 to Table 8 still need amendments. We dont normally copy and paste the output from SPSS and put in the Table here. I suggest to report only important information, such as Mean (SD), effect size, significant only. Please see other journal in PLOS one on the Table reporting. Thank you.

7. PLOS authors have the option to publish the peer review history of their article (what does this mean?). If published, this will include your full peer review and any attached files.

Reviewer #3: No

Reviewer #4: No

---

## [Author Response · Author response to Decision Letter 2]

30 Jan 2023

Presentations of Tables 3-8 have been changed. The tables have been revised and are no longer containing information on SE. We have also put confidence intervals for B in the same column as B to simplify the information. However, we believe that it is much more informative and more ‘modern’ to report the results of all variables in the table, even those that are not significant. An expert of statistical analysis has been consulted during the process. 

Some minor revisions have been made in the reference list, as you can see in the revised manuscript: One reference (Bourmans et al) has been removed since it was duplicated, therefore the reference numbers also changed throughout the manuscript. 

We have revised the manuscript on page 19 as required regarding univariate analysis revealing associations for education and psycosocail work environment.

---

## [Editor Report · Decision Letter 3]

1 Feb 2023

Individual determinants of satisfaction with the work environment after relocation to activity-based workplaces: a prospective study

PONE-D-21-33071R3

Dear Dr. Wijk,

We’re pleased to inform you that your manuscript has been judged scientifically suitable for publication and will be formally accepted for publication once it meets all outstanding technical requirements.

Kind regards,

Keiko Nakamura

Academic Editor

PLOS ONE
---

## [Editor Report · Acceptance letter]

16 Feb 2023

PONE-D-21-33071R3 

Individual determinants of satisfaction with the work environment after relocation to activity-based workplaces: a prospective study 

Dear Dr. Wijk:

I'm pleased to inform you that your manuscript has been deemed suitable for publication in PLOS ONE. Congratulations! Your manuscript is now with our production department. 

Kind regards, 

on behalf of

Professor Keiko Nakamura 

Academic Editor

PLOS ONE